# Morphological and functional cardiac alterations in children with congenital Zika syndrome and severe neurological deficits

**Imara Correia de Queiroz Barbosa**[1,2]*, **Luizabel de Paula Gomes**[1], **Israel Nilton de Almeida Feitosa**[1], **Luís Fábio Barbosa Botelho**[2,3], **Bruno Robalinho Cavalcanti Barbosa**[2], **Alex Barbosa**[1], **André Telis de Vilela Araújo**[3], **Marcelo Dantas Tavares de Melo**[3], **Adriana Suely de Oliveira Melo**[1,4], **Vera Maria Cury Salemi**[2,5]

1 Federal University of Campina Grande, Campina Grande, Brazil, 2 Heart Institute (InCor) do Hospital das Clínicas da Faculdade de Medicina da Universidade de São Paulo, São Paulo, Brazil, 3 Federal University of Paraíba, João Pessoa, Brazil, 4 Instituto de Pesquisa Professor Joaquim Amorim Neto (IPESQ), Campina Grande, Brazil, 5 Sirio Libanes Hospital, São Paulo, Brazil

* queiroz.imara@gmail.com

## Abstract

### Introduction

Zika virus infection during pregnancy causes fetal microcephaly and brain damage. Congenital Zika syndrome (CZS) is characterized by systemic involvement with diffuse muscle impairment, a high frequency of arthrogryposis, and microphthalmia. Cardiac impairment in CZS has rarely been evaluated. Our study assessed morphology and biventricular cardiac function in children with CZS and advanced neurological dysfunction.

### Methods

This cross-sectional study was conducted on 52 children with CZS (Zika group; ZG) and 25 healthy children (control group; CG) in Paraiba, Brazil. Clinical evaluation, electrocardiogram (EKG), and transthoracic echocardiogram (TTE) were performed on all children. Additionally, troponin I and natriuretic peptide type B (BNP) levels, the degree of cerebral palsy, and neuroimaging findings were assessed in the ZG group.

### Results

The median age of the study population was 5 years in both groups, and 40.4% (ZG) and 60% (CG) were female. The most prevalent electrocardiographic alteration was sinus arrhythmia in both the ZG (n = 9, 17.3%) and CG (n = 4, 16%). The morphological parameters adjusted for Z score were as follows: left ventricular (LV) end-diastolic diameter in ZG: -2.36 [-5.10, 2.63] vs. CG: -1.07 [-3.43, 0.61], p<0.001); ascending aorta (ZG: -0.09 [-2.08, 1.60] vs. CG: 0.43 [-1.47, 2.2], p = 0.021); basal diameter of the right ventricle (RV) (ZG: -2.34 [-4.90, 0.97] vs. CG: -0.96 [-2.21, 0.40], p<0.01); and pulmonary artery dimension (ZG: -2.13 [-5.99, 0.98] vs. CG: -0.24 [-2.53, 0.59], p<0.01). The ejection fractions (%) were 65.7 and 65.6 in the ZG and CG, respectively (p = 0.968). The left atrium volume indices

**Data Availability Statement:** All relevant data are in the manuscript and its supporting information files.

**Funding:** The author(s) received no specific funding for this work.

**Competing interests:** The authors have declared that no competing interests exist.

(mL/m$^2$) in the ZG and CG were 13.15 [6.80, 18.00] and 18.80 [5.90, 25.30] (p<0.01), respectively, and the right atrium volume indices (mL/m$^2$) were 10.10 [4.90, 15.30] and 15.80 [4.10, 24.80] (p<0.01). The functional findings adjusted for Z score were as follows: lateral systolic excursion of the mitral annular plane (MAPSE) (ZG: 0.36 [-2.79, 4.71] vs. CG: 1.79 [-0.93, 4.5], p = 0.001); tricuspid annular plane systolic excursion (TAPSE) (ZG: -2.43 [-5.47, 5.09] vs. CG: 0.07 [-1.98, 3.64], p<0.001); and the S' of the RV (ZG: 1.20 [3.35, 2.90] vs. CG: -0.20 [-2.15, 1.50], p = 0.0121). No differences in biventricular strain measurements were observed between the groups. Troponin I and BNP levels were normal in in the ZG. Grade V cerebral palsy and subcortical calcification were found in 88.6% and 97.22% of children in the ZG group, respectively.

## Conclusion

A reduction in cardiac dimensions and functional changes were found in CZS patients, based on the TAPSE, S' of the RV, and MAPSE, suggesting the importance of cardiac evaluation and follow-up in this group of patients.

## Author summary

Congenital Zika syndrome (CZS) occurs in the fetuses of mothers infected with the Zika virus during pregnancy, especially in the third trimester. The main findings are microcephaly and other neurological defects; however, several systemic effects have also been reported. Cardiac manifestations have rarely been evaluated. In this study, we compared cardiac anatomy and function between 52 children with CZS and 25 healthy children using electrocardiograms, echocardiograms, and biomarkers. We also assessed the degree of cerebral palsy and neuroimaging findings in the affected group. The hearts of the CZS patients were smaller and demonstrated functional alterations in comparison with those of the control individuals. Neurological impairment in the CZS patients was severe. The early detection of cardiac alterations highlights the need for cardiac evaluation and follow-up in this patient group.

## Introduction

Zika virus (ZIKV) is mainly transmitted by the *Aedes aegypti* mosquito in tropical regions [1,2]. In 1947, the Zika virus was discovered in the Zika forest in Uganda [1,3], and the first cases of human infection were detected in 1952 in Uganda and Tanzania [3–5]; however, it was only in 2013 in French Polynesia that severe neurological complications were first described, and the incidence of Guillain–Barré syndrome was observed to increase approximately 20-fold that year [2,6].

In 2015, an epidemic of ZIKV infection was declared in Brazil [7,8]. The symptoms in infected adults were mild, but in the fetuses of women infected during pregnancy, especially in the late first trimester, the disease was extremely severe, with the main alterations of microcephaly and other types of brain damage observed. The disease in fetuses and infants has been called congenital Zika syndrome (CZS), a term proposed because the pathological effects are not limited to microcephaly; indeed, other manifestations of neurological and systemic development [9], such as cortical development reduction, calcifications, ventriculomegaly,

abnormalities of the corpus callosum, microencephaly defined as reduced brain volume, and microcephaly, defined as reduced head circumference, have been observed [9–12]. In addition to brain impairment, congenital arthrogryposis, microphthalmia, dysphagia, and neuromuscular alterations are the most common systemic manifestations. [13,14]

In children with congenital Zika syndrome, the prevalence of congenital heart disease was higher than the expected rate for a population younger than 12 months of age; however, most of the defects found were hemodynamically insignificant. Atrial septal defects (ASDs), ventricular septal defects (VSDs), persistent foramen ovale, anomalous drainage of the pulmonary veins, total atrioventricular septal defects, and persistent ductus arteriosus have been described [15–17]. However, electrocardiographic, comprehensive echocardiographic and biomarker evaluations have not been evaluated in this patient population. Parte superior do formulário.

## Methodology

### Ethics statement

The research was approved by the Ethics Committee of the Federal University of Paraíba, number 4683726. Formal written consent was obtained from the parent/guardian.

Between 2021 and 2022, we recruited 52 children whose mothers were infected with ZIKV during pregnancy who were diagnosed with CZS and receiving follow-up care at the Joaquim Amorim Neto Research Institute (IPESQ) in Campina Grande, Paraíba, Brazil for our cross-sectional study. The CZS diagnosis was made in fetuses or newborns by rRT-PCR of amniotic fluid, serum, or urine or by brain imaging examination via computer tomography (CT) or magnetic resonance imaging (MRI).

The control group comprised 25 healthy children of the same age and gender range from public schools in Campina Grande, Paraíba, and the invitation to participate was extended during school meetings. Patients with congenital heart disease with hemodynamic repercussions and patients whose legal guardians did not agree to participate were excluded.

The sample was convenience-based, including all patients with CZS under follow-up at the research center in 2021. There was limited adherence in the control group since data collection occurred during the COVID-19 pandemic. Considering that the values would be adjusted by the Z score and reference values are well defined in the literature, we chose to maintain a 2:1 ratio. This study formed the doctoral thesis of the first author, Imara Correia de Queiroz Barbosa, MD, who was a student at the Heart Institute (InCor), University of São Paulo Medical School.

Physical examinations, 12-lead electrocardiograms (EKG), and transthoracic echocardiograms (TTE) were performed on all patients. Natriuretic peptide type B (BNP) and troponin levels were measured, and neuroimaging analysis was performed on the CZS group.

EKG was performed using a Wincardio device (Micromed, Brazil). Heart rhythm, heart rate, presence of chamber overloads, conduction blocks, and arrhythmias were assessed. TTE was performed on an Affiniti 50 device (Philips, Amsterdam) equipped with second harmonic imaging. The images were acquired and analyzed according to the recommendations of the American Society of Echocardiography. [18]

Linear measurements of the left atrium (LA), left ventricle (LV), right ventricle (RV), aorta, pulmonary artery, and inferior vena cava were performed and normalized according to age and body surface area using the Z score. The left and right atrium (RA) volumes were adjusted for body surface area (volume index). The ejection fraction (EF) was estimated by the biplane Simpson's method. The measurements of lateral systolic excursion of the mitral annular plane (MAPSE) and tricuspid annular plane systolic excursion (TAPSE) were performed using the M-mode and adjusted according to the Z score. Transmitral flow is sampled from the apical

four-chamber view using pulsed wave Doppler imaging. We used tissue Doppler imaging at the lateral mitral and tricuspid annulus regions to calculate peak systolic (s'), early diastolic (e'), and late diastolic (a') velocities. In patients with tricuspid regurgitation, continuous wave Doppler was used to measure the maximal velocity of tricuspid regurgitation and calculate pulmonary artery systolic pressure (PASP). The measurements of MAPSE, TAPSE, and S' from the RV were also normalized according to age and body surface area using the Z score.

For myocardial deformation index evaluation, the imaging acquisition was set to a frame rate between 40 and 80 frames per second, and it was ensured that the EKG signal was stable and that the heart rate was constant. LV global longitudinal deformation was analyzed by obtaining apical views of three chambers, four chambers, and two chambers to analyze 18 segments. LV circumferential deformation was analyzed using images acquired in the short-axis view of the parasternal window. Three dynamic images were acquired: basal at the level of the mitral valve, mid at the level of the papillary muscles, and apical for analysis of 18 segments. To investigate RV deformation, a four-chamber apical view was acquired to analyze the 3 segments of the free wall. By convention, the negative sign was not used for the global longitudinal and circumferential index evaluation value.

High-sensitivity troponin I and BNP levels were measured. Blood samples were collected on the same day as the transthoracic echocardiogram from patients with CZS. The normal reference value for BNP was <35 pg/mL since they were outpatients, and that for high-sensitivity troponin I was <0.2 ng/mL.

All patients with cerebral palsy underwent CT and MRI analysis of the brain, which was previously performed by an experienced radiologist. The level of cerebral palsy was calculated using the Gross Motor Function Classification System (GMFCS), which objectively classifies motor function based on initial movement, such as sitting, standing up, and moving around [19].

## Statistical analyses

Since the numeric variables were not normally distributed, they were described using medians and ranges, and categorical variables were described using relative and absolute frequencies. Mann–Whitney U or Kruskal–Wallis tests were used to assess differences between groups, and Fisher's exact test was used to test for differences in categorical variables. We used a linear regression model for the multivariate analysis. The interobserver analysis was performed on 20% of samples using the intraclass correlation coefficient (ICC), with values between 0.75 and 1 considered excellent. Statistical analyses were conducted using SPSS software version 23.0 (SPSS Inc. USA). The significance level was set at 5%.

## Results

We initially evaluated 53 patients in the CZS group (ZG). Unfortunately, the difficult acoustic window during the echocardiogram of one patient resulted in low-quality images, and this patient was subsequently excluded from our analysis. None of the remaining patients were excluded due to congenital heart disease with hemodynamic repercussions.

As a result, our final sample consisted of 52 patients with CZS (ZG) and 25 patients in the control group (CG). Importantly, none of the patients exhibited the clinical manifestations of heart failure or severe heart disease. Detailed characteristics of the study groups are presented in Table 1.

All patients underwent EKG. In the ZG, 9 (17.30%) patients had sinus arrhythmia, 4 (7.69%) had sinus tachycardia, 1 (1.92%) had sinus bradycardia, 1 (1.92%) had RV overload, and another (1.92%) had LV overload. In the CG, 4 (16%) patients had sinus arrhythmia, 1

**Table 1. Clinical and demographic characteristics of the study groups.**

| Variable | Control group (CG) N 25 | CZS group (ZG) N 52 | p value |
|---|---|---|---|
| Female | 15 (60%) | 21 (40.4%) | 0.144* |
| Age (years) | 5.00 [3.00, 8.00] | 5,00 [3.00, 6.00] | <0.01** |
| Height (cm) | 115 (95, 135) | 105(90, 128) | <0.01** |
| Weight (kg) | 21.15 [14.50, 31.95] | 16.00 [9.20, 28.35] | <0.001** |
| Body surface (m$^2$) | 0.86 [0.60, 1,07] | 0.66 [0,51, 0,90] | <0.01** |
| SBP (mmHg) | 104 [80, 134] | 105 [94, 118] | 0.133** |
| DBP (mmHg) | 64 [50, 80] | 70 [58, 82] | 0.045** |
| HR (bpm) | 82 [32, 106] | 90.5 [60, 120] | 0.065** |

Notes

Numeric variables are presented as medians and ranges.

Body surface area was calculated using the Dubois formula (BSA = 0.007184 * Height0.725 * Weight0.425).

CG: control group, DBP: diastolic blood pressure, HR: heart rate, SBP: systolic blood pressure, ZG: congenital Zika syndrome group.

Fisher's exact test*, Mann–Whitney tests**, P<0,05

(4%) had a right bundle branch block, and 1 (4%) had frequent monomorphic ventricular extrasystoles. The electrocardiographic parameters are listed in Table 2.

The following structural cardiac alterations were indicated by echocardiography: 46 patients had some degree of tricuspid regurgitation, allowing estimation of PSAP, including 29 (55.76%) in the ZG and 17 (68%) in the CG, Most patients had only physiological escape, whereas 9 (17.3%) in the ZG and 4 (16%) in the CG had mild tricuspid regurgitation, 2 (3.84%) in the ZG had a patent foramen ovale, 1 (1.92%) in the ZG had pulmonary artery hypoplasia, 1 (1.92%) in the ZG had mitral valve prolapse, 1 (1.92%) in the ZG had pericardial thickening, and 1 (4%) in the CG had patent ductus arteriosus with a mild shunt.

Table 3 presents the echocardiographic measurements of the left heart of patients in the two groups. Table 4 presents the echocardiographic measurements of the right heart of patients in the two groups.

We employed a linear regression model to analyze the relationships among confounding variables. Having The fact that the patient had CZS, after adjusting for sex, age, weight, and

**Table 2. Electrocardiographic parameters in both study groups.**

| Variable | Control group (CG) N 25 | CZS group (ZG) N 52 | p value |
|---|---|---|---|
| P-wave duration (ms) | 80 [40, 100] | 40 [30, 70] | <0.001 |
| QRS duration (ms) | 80 [60, 100] | 80 [50, 80] | 0.001 |
| QRS axis (°) | 60 [15, 90] | 60 [−30, 120] | 0.386 |
| QTc interval (ms) | 369 [331.5, 405.5] | 363.50 [293, 410] | 0.133 |
| R-wave amplitude V1 (mm) | 6 [0, 10] | 6 [1, 13] | 0.844 |
| R-wave amplitude V6 (mm) | 9 [5, 21] | 7 [2, 18] | 0.009 |
| S-wave amplitude V1 (mm) | 8 [2, 20] | 8 [1, 20] | 0.930 |
| S-wave amplitude V6 (mm) | 0 [0, 3] | 0 [0, 2] | 0.230 |

CG: control group; ZG: congenital Zika syndrome group

QTc: rate-corrected QT interval

Mann–Whitney tests, P<0,05

**Table 3. Echocardiographic measurements of the left heart in patients in the control and congenital Zika syndrome groups.**

| Variable | CG (n = 25) | ZG (n = 52) | p value |
|---|---|---|---|
| DdLV | -1.07 [-3.43, 0.61] | -2.36 [-5.10, 2.63] | <0.001 |
| DsLV | -0.06 [-1.98, 2.28] | -1.02 [-4.49, 0.62] | <0.01 |
| VS | -2.18 [-3.70, -0.84] | -2.26 [-3.97, -0.80] | 0.765 |
| PW | -0.99 [-2.23, 0.05] | -1.33 [-2.82, -0.15] | 0.183 |
| EF (%) | 65.6 [58.3, 71.10] | 65.70 [57.60, 75.20] | 0.968 |
| Aortic root | -0.07 [-1.67, 4.93] | 0.17 [-1.77, 3.87] | 0.349 |
| Ascending aorta | 0.43 [-1.47, 2.2] | -0.09 [-2.08, 1.60] | 0.021 |
| LA volume Index (ml/m$^2$) | 18,80 [5,90, 25,30] | 13,15 [6,80, 18,00] | <0.01 |
| MAPSE Lateral | 1.79 [-0.93, 4.50] | 0,36 [-2.79, 4.71] | 0.001 |
| E wave (cm/s) | 102 [67, 139] | 99 [60, 118] | 0.469 |
| A wave (cm/s) | 52 [26, 94] | 64 [24, 134] | 0.002 |
| E/A Ratio | 1.85 [1,37, 4,35] | 1.44 [0.76, 4.46] | <0.01 |
| Septal e' wave (cm/s) | 17 [12, 27] | 15 [7, 23] | 0.001 |
| Lateral e' wave (cm/s) | 12 [9, 15] | 11 [7, 19] | 0.016 |
| Mean e' septal and lateral (cm/s) | 15 [11, 21] | 13 [8, 21] | 0.02 |
| E/e' Ratio | 6.41 [4.75, 12.64] | 7,52 [4.57, 10.78] | <0.01 |
| Longitudinal global *Strain* | 22.30% [19.90 a 24.40%] | 21.50% [15.60 a 30.30%] | 0.263 |
| Circumferential global *strain* | 31.90% [24.20 a 39.6%] | 34.35% [15.80 a 45.40%] | 0.029 |

Measurements of the LV and aorta were corrected by Z score. DdLV: Left ventricular end-diastolic diameter, DsLV: Left ventricular end-systolic diameter, EF: Ejection fraction, MAPSE: Mitral annular plane systolic excursion, PW: Posterior wall, vs.: Ventricular septum.

Mann–Whitney tests, P<0,05

height, negatively affected the left atrial volume index (p<0.01), right atrial volume index (p<0.01), left ventricular end-diastolic diameter (p<0.01), basal right ventricular diameter (p<0.01), S' velocity of the right ventricular lateral wall (p = 0.09), tricuspid annular plane systolic excursion (p<0.01), and mitral annular plane systolic excursion (p<0.01) and had no effect on the ejection fraction (P = 0.354).

**Table 4. Echocardiographic measurements of the right heart in the control and congenital Zika syndrome groups.**

| Variable | CG (n = 25) | ZG (n = 52) | p value |
|---|---|---|---|
| Basal diameter RV | -0.96 [-2.21, 0.40] | -2.34 [-4.90, 0.97] | <0,01 |
| PA | -0.24 [-2.53, 0.59] | -2.13 [-5.99, 0.98] | <0.01 |
| RPA | -0.61 [-2.39, 1.68] | -1.47 [-3.81, 1.28] | <0.01 |
| LPA | -0.21 [-3.05, 0.94] | -1.07 [-3.45, 0.75] | <0.01 |
| TAPSE | 0.07 [-1.98, 3.64] | -2.43 [-5.47, 5.09] | <0.001 |
| S'–RV | -0.20 [-2.15, 1.50] | -1.20 [3.35, 2.90] | 0.012 |
| PASP (mmHg) | 20.00 [16.00, 24.00] | 20.00 [14.00, 34.00] | 0.740 |
| Tricuspid regurgitation velocity (m/s) | 1.98 [1.64, 2.29] | 2.00 [1.37, 2.77] | 0.891 |
| RA volume index (ml/m$^2$) | 15.80 [4.10, 24.80] | 10.10 [4.90, 15.30] | <0.01 |
| longitudinal RV *strain* | 25.43% [20.0 a 32.05%] | 26.93% [16.53 a 46.73%] | 0.220 |

Measurements of the RV, pulmonary artery, TAPSE, and S' of RV were corrected by Z score. LPA: left pulmonary artery, PA: pulmonary artery, PASP: pulmonary artery systolic pressure, RPA: right pulmonary artery, TAPSE: tricuspid annular plane systolic excursion, TRV: tricuspid regurgitation velocity.

Mann–Whitney tests, P<0,05

Of the 52 ZG patients, only 1 did not have measurements of BNP and troponin I performed on the day of the echocardiogram. All other patients had ultrasensitive troponin I levels < 0.2 ng/mL and BNP levels < 17 pg/mL.

Microcephaly was present in 36 patients in the ZG group (69.2%); however, among them, 6 were born with a normal head circumference and developed microcephaly as they grew. All patients exhibited neuroimaging alterations consistent with CZS.

The degree of cerebral palsy was evaluated in the ZG group, and the distribution of patients by degree of paralysis was grade I—2 (3.8%), grade II—1 (1.9%), grade III—1 (1.9%), grade IV —2 (3.8%), and grade V—46 (88.6%).

TC and/or cerebral MRI were evaluated in 44 out of 52 ZG patients, and the following findings were obtained, in order of prevalence: 43 (97.22%) patients had subcortical calcification; 42 (95.45%) had simplification of the gyral pattern; 41 (93.18%) had ventriculomegaly; 33 (75%) had corpus callosum hypoplasia; 27 (61.36%) had basal ganglia calcification; 11 (25%) had severe cerebellar vermis hypoplasia; 9 (20.45%) had ventricular asymmetry; 7 (15.90%) had segmental hypoplasia of the brainstem; 4 (9.09%) had asymmetric cerebellar hypoplasia; 3 (6.81%) had cerebellar calcification; 3 (6.81%) had brainstem hypoplasia; 3 (6.81%) had skull collapse; 2 (4.5%) had asymmetric cerebellar hypoplasia; 1 (2.27%) had agenesis of the corpus callosum; 1 (2.27%) had severe cerebellar vermis hypoplasia; and 1 (2.27%) had calcification in the brainstem.

## Discussion

This was the first study to evaluate cardiac alterations, including an evaluation of the ventricular myocardial deformation index, in young children with CZS. Few studies have evaluated the heart in this patient demographic and instead have focused on observing congenital heart disease during the first year of life [15,16,20,21].

Our study demonstrated that the cardiac chambers of patients with CZS were smaller than those of healthy children in the control group, corrected for age and body surface area; moreover, functional alterations were observed considering TAPSE, MAPSE, and S' of RV. These findings have not been reported previously.

In the electrocardiographic analysis, we observed that the duration of the P wave and QRS complex was shorter in the ZG group, possibly due to the smaller heart dimensions of these patients. In this group, sinus arrhythmias were the most frequently observed alterations, in addition to RV overload in one patient and LV overload in another. Electrocardiographic alterations are poorly described in this population.

The echocardiographic findings indicated more strikingly that patient hearts were smaller in the ZG than in the CG, with reduced dimensions of the LV, RV, PA, and ascending aorta and reduced LA and RA volumes, even when measurements were corrected by Z score and body surface area.

Although the heart size was reduced in CZS patients, there was no difference between the groups in LV ejection fraction or myocardial deformation index evaluation, two well-established methods for evaluating pediatric myocardial function [22].

Rossi and colleagues conducted an in vitro study that characterized ZIKV infection of human fetal cardiac mesenchymal stromal cells (fcMSCs), which play important roles in both normal and pathophysiological cardiac development and function. Their findings indicate that ZIKV predominantly enters cells using the TAM-family protein tyrosine kinase receptor AXL, affecting the fcMSC differentiation process and cell death. These observations strongly suggest that heart tissue may represent another susceptible target of ZIKV infection; however, the potential cardiac-related effects of ZIKV infection have not been well characterized [23].

Another study examined ZIKV replication in human skeletal muscle myoblasts, revealing that it impairs muscle fiber differentiation. In mouse models, it was demonstrated that ZIKV targets muscle tissue both during embryonic development and after birth. This muscle infection resulted in necrotic lesions, inflammation, fiber atrophy, and reduced expression of myogenic factors, indicating that skeletal muscle is a site of viral amplification and injury that impairs muscle development. Additionally, the study suggested that ZIKV infection during myoblast differentiation could reduce the number and area of newly formed muscle fibers, potentially causing damage during fetal and neonatal muscle development [24]. The heart is potentially affected in the same way as the skeletal muscle.

Jashari et al. performed a meta-analysis in healthy children and found that the normal mean values of the longitudinal global strain of the LV ranged from -12.9% to -26.5% (mean, -20.5; 95% CI, -20.0 to -21.0%), the normal mean value of circumferential global strain of the LV ranged from -10.5% to -27.0% (mean, -22.06; 95% CI, -21.5 to -22.5%), and the values for the radial strain of the LV ranged from 24.9% to 62.1% (mean, 45.4%; 95% CI, 43.0 to 47.8%). Variations in the different normal ranges were dependent on the equipment used, the LV end-diastolic dimension, and the age of participants. Of the 28 studies analyzed, 24 involved the use of equipment from GE Healthcare [25].

In the current study, patients in both groups presented with similar global longitudinal strain of the LV values as in previous reports, while the global circumferential strain of the LV values were higher than previously reported, indicating a lower correlation of these latter data with previous studies [25].

Regarding the TAPSE measurement in the ZG corrected by the Z score, the values were below the normal limits described by studies in healthy children and lower compared to those in the CG, suggesting that the systolic function of the RV may be impaired. The adjustment of this specific measurement by the Z score factors age to a greater extent than body surface area, and we believe that the smaller size of the cavities and the smaller body surface area of CZS patients may have influenced our results [26,27].

The reproducibility of TAPSE measurements is observed to be higher than that of other echocardiographic indices for assessing RV function [28]. However, we also found a difference in the S' measurement of the RV on tissue Doppler images, which further reinforces the observation that RV cardiac function is abnormal in children with CZS.

Koestenberger et al. evaluated 860 healthy pediatric patients and found good correlation between S' of the RV and TAPSE measurements, emphasizing the need to adjust measurements by the Z score, as conducted in the current study [26].

MAPSE measurements were also lower in the ZG. This parameter remains underutilized in the pediatric population, despite being a simple, reproducible, and effective method for assessing the longitudinal systolic function of the LV, with a good correlation with the EF [29]. This was the only functional parameter of the LV that was different between the groups in our study, with lower values observed in the ZG. Due to the variation of this parameter with body surface area and age, it was necessary to adjust the measurements by the Z score; however, age is the more important predictor in this adjustment [29]. Considering that there is a large difference in body surface area between the groups and that heart size was smaller in the ZG, this may justify the result, as in the TAPSE analysis.

The parameters of diastolic function of the LV in the ZG, although altered compared to those in the CG, still fell within the normal range for the age group studied; these parameters vary widely in children. Diastolic evaluation in the pediatric population is still based on the extrapolation of guidelines from the adult population [30].

Our echocardiographic diagnoses are compatible with physiological or mild functional alterations. None of the patients in the sample had moderate or severe heart disease dysfunction, which was previously demonstrated in the literature on the CZS population.

Di Cavalcanti et al. analyzed transthoracic echocardiograms performed in 103 children with possible CZS (mean age: 58 days) and showed that the prevalence of congenital heart disease was 13.5%, almost three times higher than expected for a population under one year of age; however, the types of alterations observed were septal defects, mostly without hemodynamic repercussions [15]. Another study evaluated 120 children with a mean age of 97 days and revealed similar findings [16]. Angelidou et al. reported the first case of CZS accompanied by severe congenital heart disease in a patient with a single ventricle who required cardiac surgery and extracorporeal membrane oxygenation support [20].

Arrais et al. evaluated children during the first four years of life, and no cases of moderate to severe congenital heart disease were found. Transient and physiological findings were observed in 21.6% of patients, and 13.7% of children were diagnosed with mild asymptomatic congenital heart disease without hemodynamic repercussions [21].

Cardiovascular complications should be treated according to relevant national and international guidelines [31]. Considering the absence of moderate to severe heart disease in CZS patients, a previous study concluded that there is currently no recommendation for cardiac screening that differs from that already performed for the general population of the same age group [16]. Based on our results, echocardiography should be routinely performed in every patient with CZS, reinforcing the need to evaluate the RV, the most affected chamber identified in our study, although no hemodynamic repercussions were observed.

While some echocardiographic parameters showed anatomical and functional alterations, the use of the biomarkers BNP and troponin I indicated that patients in the current study had normal cardiac function. Notably, to our knowledge, no previous studies in patients with CZS have employed biomarkers despite their well-validated use in the pediatric population. The levels of high-sensitivity troponin I, BNP, and pro-BNP have good correlation with heart failure severity in children, making them useful tools in determining appropriate treatment [32,33].

Our sample consisted of patients with severe neurological deficits, advanced levels of cerebral palsy, and severe neurological compromise on neuroimaging.

Instances of critical illness in infants and children often imply a hypermetabolic state. Children with cerebral palsy commonly manifest nutritional and growth disorders, occasionally requiring gastrostomy tube feeding due to severe malnutrition, dysphagia, feeding difficulties, and the risk of broncho-aspiration [34]. Substantial nutritional deficits can impede musculature development, potentially impacting cardiac musculature. Aguiar et al. observed that between the ages of 25 and 36 months, over 50% of CZS children fell below weight and height thresholds, with growth restriction being evident since gestation [35]. The growth deficit in CZS likely results from a combination of factors, and our findings suggest that the heart may also be affected by this growth restriction. Microcephaly is one of the main features associated with CZSl however, not all affected patients present with this anomaly. In our study, some patients had normal head circumference but still showed characteristic disease-related changes on neuroimaging, such as the presence of subcortical calcifications. A similar finding was also observed by Aragão et al. when evaluating 19 patients with CZS, and this type of calcification was found in all cases without microcephaly [36].

Subcortical calcification was the most frequent neuroimaging finding in our study population, which is consistent with the literature [37–39]. Simplification of the gyral pattern and ventriculomegaly were frequently identified in the patients in our study. In a case series of fetuses and neonates with suspected ZIKV-related congenital syndrome conducted in

Pernambuco, ventriculomegaly was present in all cases [40], and in another case series with patients from Ceará, ventriculomegaly and alterations in the gyral pattern were common findings [41].

Almost all patients with ZIKV-related congenital syndrome in our sample had level V cerebral palsy. Even among those without microcephaly, level V was found in most patients. This finding is consistent with the literature since the absence of microcephaly does not exclude severe neurological impairment [10,42].

Paixão et al. conducted a population-based cohort study to estimate mortality among live-born children with CZS compared with that in those without the syndrome and identified cerebral palsy as the leading cause of death in both groups of children under 1 year and under 3 years of age [43].

This study has some limitations worth considering. The function of the RV can be estimated by TAPSE, S' velocity on tissue Doppler and by calculating the fractional area change (FAC). The latter method has good correlation with the EF assessed by MRI (53); however, its implementation is limited by the difficulty associated with obtaining an excellent echogenic window in the cardiac apex, which is necessary for the method. Additionally, the COVID-19 pandemic limited the ability of patients residing in other states to access the IPESQ research center and made it difficult for parents to allow the children in the control group to attend medical clinics.

In conclusion, although congenital Zika syndrome is a rare condition, its impact is profound, causing serious neurological damage and affecting other organs. This study reveals a concerning reduction in the size and functional changes of the heart. It is crucial to conduct further research to better understand how the heart is affected, whether it is due to direct viral action or nutritional deficits. Early identification of cardiac anomalies underscores the importance of conducting regular cardiac assessments and providing continuous follow-up for this group of patients, who often face vulnerabilities and heavily rely on access to public healthcare services.

## Supporting information

**S1 Table. Excel spreadsheet containing the individual parameters that generated Table 1.** (XLSX)

**S2 Table. Excel spreadsheet containing the individual parameters of the electrocardiographic analysis that generated Table 2.** (XLSX)

**S3 Table. Excel spreadsheet containing the individual parameters of the echocardiographic analysis that generated Table 3.** (XLSX)

**S4 Table. Excel spreadsheet containing the individual parameters of the echocardiographic analysis that generated Table 4.** (XLSX)

## Author Contributions

**Conceptualization:** Alex Barbosa, André Telis de Vilela Araújo, Adriana Suely de Oliveira Melo.

**Data curation:** Luizabel de Paula Gomes.

**Formal analysis:** Luís Fábio Barbosa Botelho.

**Investigation:** Imara Correia de Queiroz Barbosa, Luizabel de Paula Gomes, Bruno Robalinho Cavalcanti Barbosa.

**Methodology:** Marcelo Dantas Tavares de Melo, Adriana Suely de Oliveira Melo.

**Project administration:** Imara Correia de Queiroz Barbosa.

**Supervision:** Adriana Suely de Oliveira Melo, Vera Maria Cury Salemi.

**Validation:** Israel Nilton de Almeida Feitosa.

**Visualization:** Imara Correia de Queiroz Barbosa.

**Writing – original draft:** Imara Correia de Queiroz Barbosa, Bruno Robalinho Cavalcanti Barbosa.

**Writing – review & editing:** Alex Barbosa, Marcelo Dantas Tavares de Melo, Vera Maria Cury Salemi.

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
