## [Decision Letter · Decision Letter 0]

21 Aug 2023

Dear Mrs Barbosa,

Thank you very much for submitting your manuscript "Morphological and functional cardiac alterations in children with congenital Zika syndrome and severe neurological deficits Cardiac alterations in children with congenital Zika syndrome" for consideration at PLOS Neglected Tropical Diseases. As with all papers reviewed by the journal, your manuscript was reviewed by members of the editorial board and by several independent reviewers. In light of the reviews (below this email), we would like to invite the resubmission of a significantly-revised version that takes into account the reviewers' comments. 

We cannot make any decision about publication until we have seen the revised manuscript and your response to the reviewers' comments. Your revised manuscript is also likely to be sent to reviewers for further evaluation.

Sincerely,

Andrés F. Henao-Martínez, M.D.

Academic Editor

Andrea Marzi

Section Editor

Reviewer's Responses to Questions

**Key Review Criteria Required for Acceptance?**

**Methods**

-Are the objectives of the study clearly articulated with a clear testable hypothesis stated?

-Is the study design appropriate to address the stated objectives?

-Is the population clearly described and appropriate for the hypothesis being tested?

-Is the sample size sufficient to ensure adequate power to address the hypothesis being tested?

-Were correct statistical analysis used to support conclusions?

-Are there concerns about ethical or regulatory requirements being met?

Reviewer #1: For Methods of this study, my primary concerns are: 1) the determination/consideration of the study's sample size or power seems have not been delineated, and 2) the current manuscript only used bivariate analysis to access the epidemiological associations, and extra multivariate analyses adjusting for confounders are needed.

Reviewer #2: -Are the objectives of the study clearly articulated with a clear testable hypothesis stated?

Yes

-Is the study design appropriate to address the stated objectives?

Yes

-Is the population clearly described and appropriate for the hypothesis being tested?

The population description for the CZS group is well described, however there are very few details of the healthy group, I would suggest that they do a better description of the population and specify if they applied any selection criteria.

-Is the sample size sufficient to ensure adequate power to address the hypothesis being tested?

I believe that the 2:1 ratio of cases and controls should be reviewed. If there is any reason why they made the decision of this relationship, it would be interesting if they describe it in the discussion, since for the results achieved I do not believe that the sample size is adequate. I understand the characteristics of the study population but it would be nice to increase the controls. If there is any support for that decision, ignore this recommendation.

-Were correct statistical analysis used to support conclusions?

Yes

-Are there concerns about ethical or regulatory requirements being met?

No

Reviewer #3: The methods section needs to be rewritten as there are several repetitions. Also, Line 127 to 129 is an irrelevant information from the perspective of the manuscript.

**Results**

-Does the analysis presented match the analysis plan?

-Are the results clearly and completely presented?

-Are the figures (Tables, Images) of sufficient quality for clarity?

Reviewer #1: For Results of this study, things that can be improved are: 1) the footnotes in Tables can be strengthened, 2) when prevalence estimate of a health outcome is presented, the author can consider providing its 95% confidence interval to indicate uncertainty, and 3) some prevalence estimates mentioned in point 2) may be summarized in new tables to improve readability.

Reviewer #2: -Does the analysis presented match the analysis plan?

Yes

-Are the results clearly and completely presented?

Yes

-Are the figures (Tables, Images) of sufficient quality for clarity?

Yes

Reviewer #3: The study Morphological and functional cardiac alterations in children with congenital Zika

syndrome and severe neurological deficits is an interesting one and describes the first to evaluate cardiac alterations, including an evaluation of the ventricular myocardial deformation index, in young children with CZS. In the opinion of this reviewer, the work is clinically relevant and definitely shades light on the pathologies associated with CZS. However, this is a purely clinical work that was done in a single centre. While the clinical data is unequivocal and thoroughly done, the authors do not offer any mechanistic explanations to the same. Such investigations will be important to better understand the disease and also to identify therapeutic approaches. Could the authors attempt to provide such explanations?

**Conclusions**

-Are the conclusions supported by the data presented?

-Are the limitations of analysis clearly described?

-Do the authors discuss how these data can be helpful to advance our understanding of the topic under study?

-Is public health relevance addressed?

Reviewer #1: For Conclusions, the public health implications for the population at large seems have not been mentioned in the manuscript. Rather, the current conclusion paragraph seems focus solely on the next steps for the study population.

As mentioned in my comments for Methods, the conclusions need to be further corroborated by extra multivariate analysis.

Reviewer #2: -Are the conclusions supported by the data presented?

Yes, but They could improve the recommendation of the line 424 - 426

-Are the limitations of analysis clearly described?

Yes

-Do the authors discuss how these data can be helpful to advance our understanding of the topic under study?

Yes. I have a recommendation in this regard, in line 279 - 283, it mentions one of the strengths of this study that presents original and novel results, however it would be important to mention that in vitro studies have been carried out, you can review the following article, it may help you as an important precedent for its results. 

Rossi F, Josey B, Sayitoglu EC, Potens R, Sultu T, Duru AD, Beljanski V. Characterization of zika virus infection of human fetal cardiac mesenchymal stromal cells. PLoS One. 2020 Sep 17;15(9):e0239238. doi: 10.1371/journal.pone.0239238. Erratum in: PLoS One. 2021 Jan 22;16(1):e0246112. PMID: 32941515; PMCID: PMC7498051.

-Is public health relevance addressed?

Yes

Reviewer #3: (No Response)

**Editorial and Data Presentation Modifications?**

Reviewer #1: All editorial comments along with their corresponding highlighted areas in the manuscript are embedding in the attached document.

Reviewer #2: Minor revision

Line 121. The sentence of line 121 I think is unnecessary. The specification that they were born to mothers who were infected during pregnancy could be added to lines 116 and 117.

Reviewer #3: (No Response)

**Summary and General Comments**

Reviewer #1: Prior to being accepted for publication, this manuscript can still be significantly improved by considering the comments provided in my former responses and the attachment.

Regarding data availability, although the author mentioned that the data are available in the submitted materials, it seems no supplementary materials containing the individual-level data were available. The authors may upload these data in revision if have not done so.

Reviewer #2: This study is of great impact due to the scarcity of similar studies in vivo, it has an adequate design and the results are interesting in this study population, which has had an impact in several Latin American countries, with the greatest impact in Brazil. This type of study will allow the provision of adequate medical care at an early age to improve the health and quality of life of the population studied.

Reviewer #3: The study reports that the heart size was reduced in the CZS children but there was no difference in the LV ejection volume or myocardial deformation between this group and healthy controls. Also, none of the patients in the study had moderate or severe heart disease dysfunction. These observations raises the question of the implication of the study. Apart from advocating for the need of regular monitoring of cardiac function by echocardiography, the study does not really offer much in terms of progress of the field. Even the cardiac function biomarkers do not show any significant changes. 

The other part of the manuscript deals with the neurological abnormalities. Here also, the authors do not offer anything more in-depth than "severe neurological impairment was also observed in these patients.", which has already been reported by previous studies. In sum, the work suffers from the lack of novelty and also lack of mechanistic implications. Further long-term follow-up of the patients with the cardiac anomalies might have provided important clues to the disease pathogenesis.

PLOS authors have the option to publish the peer review history of their article (what does this mean?). If published, this will include your full peer review and any attached files.

Reviewer #1: Yes: Dehao Chen

Reviewer #2: Yes: Evelin Martinez

Reviewer #3: No
---

## [Decision Letter · Decision Letter 1]

30 Oct 2023

Dear Mrs Barbosa,

We are pleased to inform you that your manuscript 'Morphological and functional cardiac alterations in children with congenital Zika syndrome and severe neurological deficits Cardiac alterations in children with congenital Zika syndrome' has been provisionally accepted for publication in PLOS Neglected Tropical Diseases.

Best regards,

Andrés F. Henao-Martínez, M.D.

Academic Editor

Andrea Marzi

Section Editor

Reviewer's Responses to Questions

**Key Review Criteria Required for Acceptance?**

**Methods**

-Are the objectives of the study clearly articulated with a clear testable hypothesis stated?

-Is the study design appropriate to address the stated objectives?

-Is the population clearly described and appropriate for the hypothesis being tested?

-Is the sample size sufficient to ensure adequate power to address the hypothesis being tested?

-Were correct statistical analysis used to support conclusions?

-Are there concerns about ethical or regulatory requirements being met?

Reviewer #1: (No Response)

Reviewer #2: Accept

Reviewer #3: (No Response)

**Results**

-Does the analysis presented match the analysis plan?

-Are the results clearly and completely presented?

-Are the figures (Tables, Images) of sufficient quality for clarity?

Reviewer #1: (No Response)

Reviewer #2: Accept

Reviewer #3: (No Response)

**Conclusions**

-Are the conclusions supported by the data presented?

-Are the limitations of analysis clearly described?

-Do the authors discuss how these data can be helpful to advance our understanding of the topic under study?

-Is public health relevance addressed?

Reviewer #1: (No Response)

Reviewer #2: Accept

Reviewer #3: (No Response)

**Editorial and Data Presentation Modifications?**

Reviewer #1: Thanks for considering my suggestions in revision. The following are the final suggestions moving to the publication phase:

1) It seems a "Reference" heading is missed, so please ensure it is added in the manuscript proof.

2) In the uploaded "EKG" datasets, I could not find a codebook/directory explaining the meaning of each variable. Please ensure it is added for the benefit of future readers.

Reviewer #2: Accept

Reviewer #3: (No Response)

**Summary and General Comments**

Reviewer #1: (No Response)

Reviewer #2: Accept

Reviewer #3: The authors have adequately addressed the issues and have provided satisfactory explanations to the questions or doubts raised. The Methodology has been re-written to avoid repetitions. In my opinion, the manuscript can now be accepted for publication.

PLOS authors have the option to publish the peer review history of their article (what does this mean?). If published, this will include your full peer review and any attached files.

Reviewer #1: No

Reviewer #2: **Yes: **Evelin Martinez

Reviewer #3: **Yes: **Gayatri Mukherjee

---

## [Editor Report · Acceptance letter]

23 Nov 2023

Dear Mrs Barbosa,

We are delighted to inform you that your manuscript, "Morphological and functional cardiac alterations in children with congenital Zika syndrome and severe neurological deficits ," has been formally accepted for publication in PLOS Neglected Tropical Diseases.

Best regards,

Shaden Kamhawi

co-Editor-in-Chief

Paul Brindley

co-Editor-in-Chief
